# Do Not Miss Acute Diffuse Panbronchiolitis for Tree-in-Bud: Case Series of a Rare Lung Disease

**DOI:** 10.3390/diagnostics12071653

**Published:** 2022-07-07

**Authors:** Johannes Raedler, Hannes Hoelz, Anna Zschocke, Judith Loeffler-Ragg, Marco Paolini, Julia Ley-Zaporozhan, Matthias Griese

**Affiliations:** 1Dr. von Hauner Children’s Hospital, Department of Pediatrics, University Hospital, LMU Munich, 80337 Munich, Germany; johannes.raedler@med.uni-muenchen.de (J.R.); hannes.hoelz@med.uni-muenchen.de (H.H.); 2Department of Pediatric and Adolescent Medicine, Pediatrics III, Medical University, 6020 Innsbruck, Austria; anna.zschocke@tirol-kliniken.at; 3Department of Internal Medicine, Medical University, 6020 Innsbruck, Austria; judith.loeffler@i-med.ac.at; 4Department of Radiology, University Hospital, LMU Munich, 80337 Munich, Germany; marco.paolini@med.uni-muenchen.de (M.P.); julia.leyzaporozhan@med.uni-muenchen.de (J.L.-Z.); 5German Center for Lung Research (DZL), 80337 Munich, Germany

**Keywords:** case report, pediatric pulmonology, pulmonology, respiratory failure, acute bronchiolitis, acute diffuse panbronchiolitis, tree-in-bud pattern

## Abstract

Acute bronchiolitis is a common disease of infants affecting the small airways. Rarely, acute bronchiolitis may occur in adolescents and adults. Here, we present four unrelated adolescent patients with severe clinical presentation and unique CT imaging with extensive tree-in-bud pattern, representing a rare clinical phenotype of acute diffuse panbronchiolitis. This characteristic disease pattern caused by inhalation injury from waterpipes, smoked tobacco, and cannabinoids must be differentiated from e-cigarette or vaping product-use-associated lung injury (EVALI). Visual diagnosis of CT and an early diagnostic procedure for detection and differentiation of inhaled hazards, including sample storage for future identification of novel noxious agents, are warranted.

## 1. Introduction

Acute bronchiolitis in infants and young children is a common disease of the small airways. Typically, viruses or nonviral infectious agents, including *Mycoplasma* and *Chlamydophila*, cause an infection of the upper airways, which may progress to lower respiratory symptoms, including coughing, tachydyspnea, or wheezing [1,2].

Acute bronchiolitis in adolescence and adulthood is very uncommon and presents as an ill-defined illness with cough and dyspnea. Bronchiolitis involves small airways <2 mm in diameter and may be divided into constrictive/obliterative and cellular bronchiolitis. The latter is characterized by the presence of inflammatory cells and may be histopathologically differentiated into (1) infectious bronchiolitis, (2) aspiration bronchiolitis, (3) respiratory bronchiolitis (e.g., tobacco smoking), (4) hypersensitivity pneumonitis (allergic bronchiolitis), (5) follicular bronchiolitis (seen in autoimmunity), and (6) diffuse panbronchiolitis [2,3]. Diagnosis requires a lung biopsy, which is frequently omitted. If bronchial/bronchiolar wall thickening or nodular lung pattern is not apparent on conventional chest radiographs, bronchiolar disease (BD) may easily be missed. Therefore, high-resolution computed tomography (HR-CT) imaging is required. Abnormal bronchioles manifest as direct findings on HR-CT through centrilobular nodules, tree-in-bud pattern, and bronchial/bronchiolar wall thickening, or as indirect findings through mosaic attenuation due to air trapping [3]. Disorders may be divided into common and uncommon conditions, but the important clinical feature of acuity of presentation is often neglected. Most patients with BD present with subacute or chronic conditions. 

Here, we present the exceptional clinical course of four unrelated adolescents, who presented with acute respiratory failure requiring ventilation and a phenotype of acute diffuse panbronchiolitis, based on homogenously disseminated tree-in-bud pattern on CT imaging.

## 2. Case Presentations

### 2.1. Case 1

A previously healthy 17-year-old woman was admitted for progressive dry cough, dyspnea, and fever (see Table 1). She reported infrequent use of a waterpipe and cigarettes, and having tried marijuana 5 days before symptom onset. Respiratory insufficiency with hypercapnia (pCO_2_ max. 82.0 mmHg, Figure 1A) developed after a diagnostic bronchoalveolar lavage (BAL; Table 1). She was transferred to the ICU for 3 days of mechanical ventilation. Radiologic imaging revealed surprisingly unremarkable chest X-rays, but diffuse and homogenously distributed centrilobular micronodules and a tree-in-bud pattern on HR-CT (Figure 2A,B). After clinical stabilization, she was weaned within 1 week and discharged in stable condition 31 days after admission with no oxygen demand. Extensive work-up showed no causative agent, merely borderline increased serological results for Mycoplasma pneumoniae (slightly raised IgA, max. IgG 49.9 U/mL, IgM negative), without significant increases in follow-up tests. An outpatient corticosteroid burst (methylprednisolone, 8 mg/kg for 3 days) produced no significant improvement (Figure 1B). At last follow-up, 23 months after admission, Patient 1 was off medication with improving exercise tolerance and spirometry results. 

### 2.2. Case 2

A previously healthy 16-year-old male patient was admitted with an 8-day history of coughing, hemoptysis, and fever. He reported frequent marijuana and waterpipe use. Following respiratory failure, he was transferred to the ICU and received mechanical ventilation for 22 days, complicated by a right-sided pneumothorax. Chest X-ray revealed increased bilateral peribronchial and interstitial markings (Figure 2C). HR-CT revealed a diffuse and homogenously distributed tree-in-bud pattern (Figure 2D). Broad work-up revealed a positive serology for *Chlamydophila pneumoniae* (IgG 188 U/mL, IgA 170 U/mL) but no PCR detection. No further causative agents were found (Table 1). He was weaned and discharged 39 days after admission. Subsequent treatment involved ten-monthly methylprednisolone bursts with steadily improving spirometry (Figure 1B). At last follow-up, he reported smoking one cigarette daily but no illicit drug use. 

### 2.3. Case 3

A 16-year-old male patient was recently admitted to our hospital with a 5-day history of coughing, intermittent slight hemoptysis, orthostatic dizziness, and postprandial emesis. Regular nicotine and cannabis consumption and waterpipe use were reported (Table 1). The patient had a mild oxygen demand with no hypercapnia. Chest X-ray showed increased bilateral peribronchial and interstitial markings and small pleural effusions (Figure 2E). CT imaging showed a diffuse and homogenous tree-in-bud pattern (Figure 2F). Screening for infectious causes revealed positive IgG against *Chlamydophila pneumoniae* (IgG 105 U/mL, IgM and IgA negative) and *Mycoplasma pneumoniae* (IgG 31 U/mL, IgM and IgA negative; Table 1). Further diagnostics were delayed due to clinical deterioration with increased oxygen demand (max. 6 L/min) on day 5 due to secondary SARS-CoV-2 infection. Treatment included antibiotics, corticosteroids (max. 1 mg/kg/d), and ASS; no mechanical ventilation was necessary. The patient slowly recovered and was discharged on day 22. The prednisolone treatment was tapered over 4 weeks. Scheduled BAL and transbronchial lung biopsy on day 59 were postponed due to repeatedly positive SARS-CoV-2 tests. Further diagnostics and follow-up appointments were declined by the patient.

### 2.4. Case 4

A previously healthy 16-year-old male patient who regularly smoked tobacco and e-cigarettes was admitted to a peripheral hospital with an 8-day history of cough and intermittent hemoptysis. He had smoked marijuana two days before admission, and subsequently developed increasing shortness of breath and fever. Upon admission, he was initially treated for community-acquired pneumonia. A chest X-ray revealed bilateral widespread nodular opacities, and a course of IV steroids was initiated on suspicion of hypersensitivity pneumonitis. His respiratory situation continued to worsen, and the patient was transferred to the pediatric ICU after 2 days for mechanical ventilation. HR-CT revealed a diffuse and homogenously distributed tree-in-bud pattern (Figure 2H). Bronchial lavage produced nonspecific findings with 3% Lymphocytes and no infectious pathogens. Precipitating antibodies for fungus, bird feathers, and droppings were negative. Under IV prednisolone (1.5 mg/kg/d), he could be weaned after 5 days, required no further supplemental oxygen, and was discharged 14 days after admission on a tapered course of steroids, with low-dose steroids planned for the following months. On the day of discharge, spirometry demonstrated a fixed obstructive ventilation disorder with no sign of a restrictive ventilation pattern (Appendix A). 

## 3. Discussion

Acute bronchiolitis is a rare condition in adolescence because the bronchioles, i.e., airways smaller than 2 mm in diameter, contribute less (<20%) to total airway resistance than they do in infants and young children (>50%) [2]. Therefore, extensive involvement of small airways must be present before clinical symptoms occur in adolescents or adults. 

Our patients presented with severe bronchiolitis, requiring mechanical ventilation in three of four patients. The CT imaging pattern was unequivocally diagnostic of BD with a tree-in-bud pattern, i.e., the presence of direct radiographic findings of abnormal bronchioles. Common features of all cases were recent usage of inhalation hazards, and similar CT pattern and clinical courses. The slightly raised serology results for *Mycoplasma pneumoniae* (Patients 1 and 3) or *Chlamydophila pneumoniae* (Patients 2 and 3) were insufficient explanations for the extensive CT findings. Additionally, no signs of vascular causes, particularly the intravenous compound injection typical for cellulose granulomatosis [3], were present. 

As a projection imaging modality, subtle variations in lung density may be under-represented in conventional chest X-rays. Therefore, the shown discrepancy between unremarkable or moderate findings on chest X-ray and extensive CT abnormalities with a diffuse and homogenously disseminated tree-in-bud pattern are noteworthy for clinicians. The most feasible underlying entity is diffuse acute panbronchiolitis, which must be differentiated from the Asian “diffuse panbronchiolitis”. The latter is a well-characterized chronic condition in adults, defined by criteria by the Japanese Ministry of Health and Welfare. It is characterized by bronchiolitis and chronic sinusitis, which typically develop slowly and progressively over months to years [4,5]. While the mean age at presentation is 40 years, Asian panbronchiolitis occasionally develops in childhood [5,6]. Chronic sinusitis was found in at least 75% of Japanese patients, often preceding chest symptoms by years [5]. With negative histories for chronic sinusitis and an acute disease onset, the diagnosis was not supported in any case [6]. Due to the lack of histological specimens and lung biopsy being infeasible during respiratory decompensation, the working diagnosis of acute diffuse panbronchiolitis was made upon clinical presentation and typical radiographic findings.

Unfortunately, our diagnostic work-up of the two initial patients was challenged by various factors: external presentation and late transfer of patients to our care, or incomplete, perhaps unreliable, patient histories with respect to the brands and providers of the novel hazardous compounds consumed by the adolescents. We immediately and visually diagnosed the recent third case, although he plausibly denied smoking habits at first. Follow-up CT imaging showed centrilobular nodular hazy densities with upper lobe predominance suspect for respiratory bronchiolitis for Patient 2 after 35 months (Figure 3B), resembling persistent exposure to inhaled noxae, while reporting smoking one cigarette daily and no illicit drug use. The follow-up CT for Patients 1 and 3 showed nearly complete resolution after 6 months and significant improvement after 2 months, respectively (Figure 3A,C). Given the more recent manifestation in Patient 4, no follow-up CT was available at the time of submission.

The emergence of new smoking systems, particularly e-cigarettes, has been followed by e-cigarette, or vaping, product-use-associated lung injury (EVALI), particularly among adolescents and young adults [7]. In addition to nicotine, e-cigarettes may be used to deliver tetrahydrocannabinol (THC) or cannabidiol (CBD) and numerous flavoring or supporting ingredients, which are not strongly regulated. Blount et al. recently identified vitamin E acetate as a possible causative agent of EVALI [8]. Only seven cases of EVALI with one fatal outcome have been reported outside of the U.S. so far [9,10,11,12,13,14]. Between the fall of 2019, when EVALI was first reported, and 18 February 2020 (the last date of data collection), there were 2807 confirmed cases in the United States requiring hospital admission and 68 deaths [15]. The use of e-cigarettes or other vaping products could not be proven in three of the four patients, excluding the diagnosis of EVALI according to Layden et al. [7]. Furthermore, EVALI typically manifests with bilateral pulmonary infiltrates on a chest X-ray, not seen in our patients [7]. Respiratory bronchiolitis must be differentiated as another inhaled drug-/smoke-induced lung injury pattern. Memorizing this pattern helps in the differential diagnostic work-up. Generally, we categorize the acute diffuse panbronchiolitis as an exposure related, non-infectious diffuse parenchymal disorder [16]. In addition to the respiratory bronchioles, the pulmonary interstitial space with the adjacent alveolar ducts and alveoli are affected.

After reviewing the literature, we identified one similar case of a 17-year-old patient with severe bronchiolitis following *Mycoplasma pneumoniae* infection [17]. However, the right-sided predominance of disease manifestation clearly visible on chest radiograph, absence of severe respiratory distress, and positive infectious correlation were key distinctions from our patients. The chest radiographic findings with comparable signs of bronchiolitis in a series of four cases after inhalation of synthetic marijuana fit the histories and clinical courses of our Patients 2 and 4 [18]. Alhadi et al. reported the clinical course of a patient who presented with diffuse pulmonary infiltrates related to chronic inhalation of multiple synthetic-cannabinoid-containing products [19]. As in our cases, the patient was treated with antibiotics and steroids. Given the rapid resolution of radiographic and clinical derangements after the administration of corticosteroids, the authors considered the differential diagnosis of hypersensitivity pneumonitis (HP), also known as exogenous allergic alveolitis (EAA). HP typically presents with a restrictive pulmonary process resulting from the patient´s hypersensitivity to bio-aerosols, whereas our patients clearly had an obstructive lung function impairment and no exposure history (Figure 1B [20]. The use of waterpipes was reported by three of four patients, another popular smoking system among adolescents, often falsely considered less dangerous through the supposed filtering effects of water [21]. 

A combination of yet unknown precipitating factors, such as undetected infection or toxins, may have been causative in our patients. To assist colleagues facing similar cases, we recommend the following diagnostic work-up:An adequate medical history is the key to proper diagnosis and therapy. Therefore, a thorough history must be taken in a calm, trust-building environment to obtain honest answers to sensitive questions regarding drug use.Early CT scans should be taken to identify BD in adolescents with potential inhalation hazards and progressive pulmonary symptoms.Broad infectious work-up including sputum, serum (blood cultures, titers, and PCR), and BAL analysis should be performed.Lung biopsy (video-assisted thoracic surgery or trans-bronchial biopsy) should be performed if clinically feasible for histological diagnosis and if uncertain about diagnosis.Initial urine and blood specimens should be taken for toxicological analysis (e.g., immunoassays, gas, or liquid-chromatography mass spectrometry). Consider that novel drugs may not be included in standard diagnostic panels.

## Figures and Tables

**Figure 1 diagnostics-12-01653-f001:**
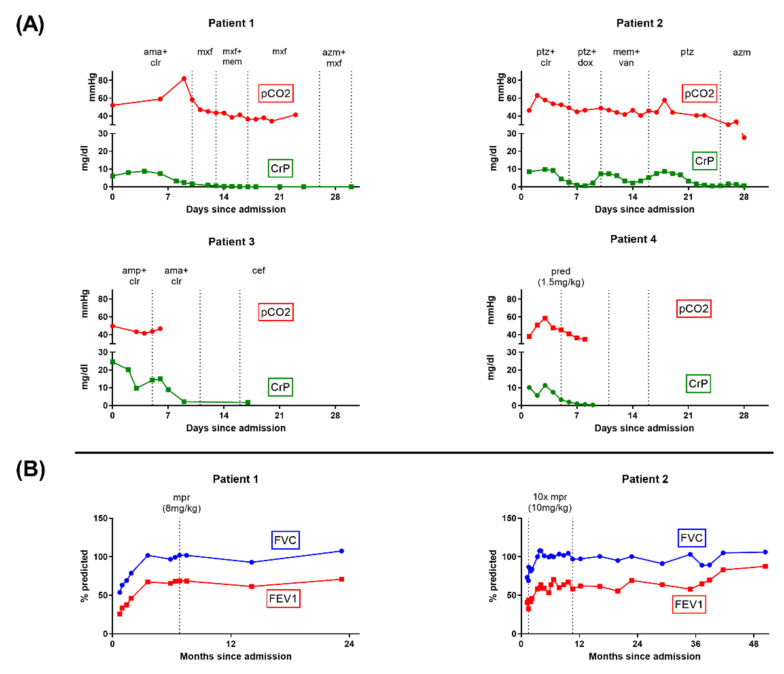
Clinical courses and spirometry results. (**A**) Results of pCO2 from blood gas analysis and C-reactive protein (CRP) from laboratory studies during hospitalization. Antibiotic treatment intervals are indicated. Additionally, the patients received systemic prednisolone for reduction in airway inflammation. All patients showed a normalization of pCO2 and CRP levels prior to discharge. (**B**) Spirometry results during follow-up and methylprednisolone burst treatments for Patients 1 and 2 (dosage indicated, each as daily dosage for 3 days). Initial spirometry showed peripheral obstruction, overinflation, increased airway resistance, and pseudorestriction in both patients. No spirometry follow-up was available for Patient 3 or 4. amp = ampicillin, ama = ampicillin/sulbactam, azm = azithromacin, cef = cefpodoxime, clr = clarithromycin, dox = doxycycline, CT = computed tomography, mem = meropenem, mpr = methylprednisolone burst, mxf = moxifloxacin, ptz = piperacillin-tazobactam, pred = prednisolone, and van = vancomycin.

**Figure 2 diagnostics-12-01653-f002:**
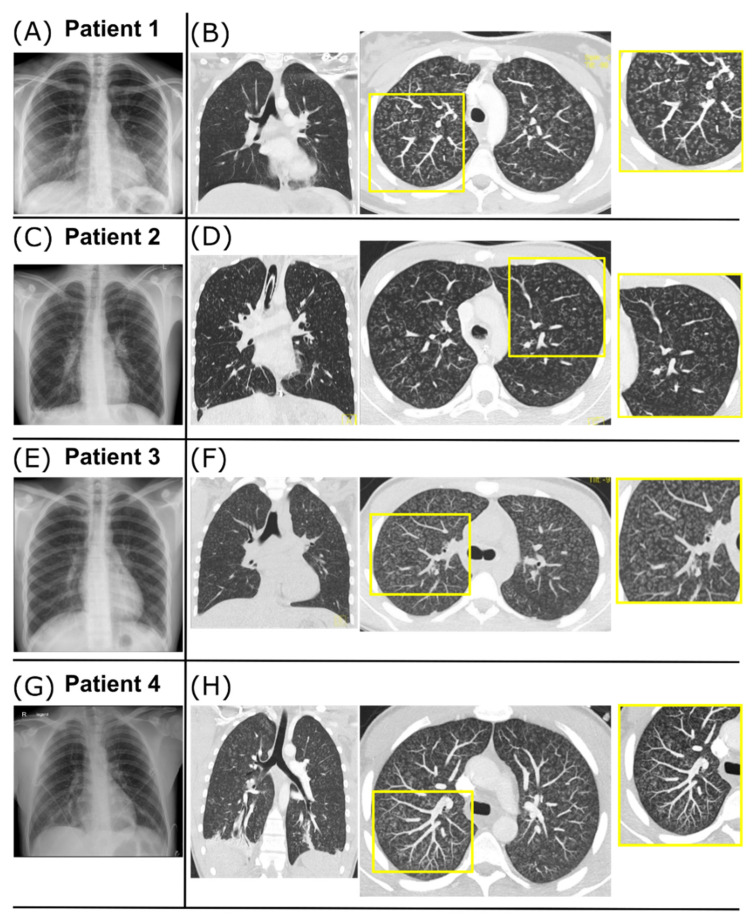
**Radiographic findings.** (**A**,**B**) Radiographic findings for Patient 1. Chest X-ray on day 2 showing signs of peribronchial cuffing (**A**). Coronal CT image on day 8 shows diffuse and homogenous distribution of centrilobular micronodules and tree-in-bud pattern through the whole lung (**B**, left). Axial maximal intensity projection (MIP, 5 mm thickness) of the CT scan improves the detection and distribution of the disease pattern (**B**, right). In the magnified view, the tree-in-bud pattern is mostly explained by peribronchial thickening. (**C**,**D**) Radiographic findings for Patient 2. Chest X-ray on day 1 shows slight enlargement of the hilar lymph nodes, increased peribronchial and interstitial markings, small opacification in the right basal area, and small bilateral pleural effusions (**F**). Coronal CT image on day 4 (**G**, left) and axial MIP magnification (**G**, right) show diffuse and homogenous tree-in-bud pattern, mostly due to endobronchial pathology (**G**). (**E**,**F**) Radiographic findings for Patient 3. Chest X-ray on day 6 shows diffuse bilateral peribronchial markings with basal dominance and small pleural effusions suspicious for atypical pneumonia (**E)**. Coronal CT image on day 6 (**F**, left) and axial MIP magnification (**F**, right) show tree-in-bud pattern due to peribronchial thickening with diffuse and homogenous distribution through the whole lung. (**G**,**H**) Radiographic findings for Patient 4 (provided by the Department of Radiology, Medical University Innsbruck, Austria). Chest X-ray on day 1 shows bilateral widespread nodular opacities (**G**). Coronal CT image on day 2 (**H**, left) and axial MIP magnification (**H**, right) show diffuse and homogenous tree-in-bud pattern through the whole lung caused by peribronchial thickening (**H**).

**Figure 3 diagnostics-12-01653-f003:**
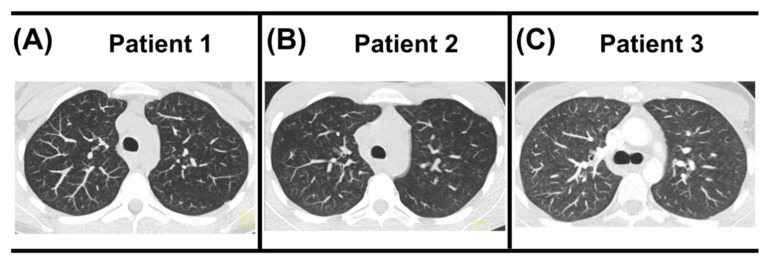
CT follow-up. (**A**) Follow-up CT of Patient 1 after 6 months showed near complete resolution of the previously extensive tree-in-bud pattern. (**B**) Follow-up CT of Patient 2 after 35 months showed no tree-in-bud pattern but centrilobular nodular hazy densities with upper lobe predominance suspicious for respiratory bronchiolitis disease, possibly secondary to inhaled noxae. (**C**) Follow-up CT of Patient 3 after two months showed significant improvement. Due to the recent manifestation, no follow-up CT of Patient 4 was currently available.

**Table 1 diagnostics-12-01653-t001:** Patient information and medical work-up.

	Patient 1	Patient 2	Patient 3	Patient 4
**Age (years)**	17	16	16	16
**Sex**	Female	Male	Male	Male
**Presentation**	Progressive dry cough, dyspnea and fever, despite 7-day clarithromycin treatment	Progressive cough, hemoptysis and fever	5-day history of coughing, intermittent slight hemoptysis, orthostatic dizziness, postprandial emesis; no fever	8-day history of coughing, intermittent hemoptysis
**Days intubated**	3	22	none	5
**Past medical history**	Uneventful	Uneventful	Uneventful	Uneventful
**Asthma or pulmonary disease**	No	No	No	No
**Allergies**	None	None	None	Grass pollen
**Vaccinations**	Fully vaccinated	Fully vaccinated	Fully vaccinated	Fully vaccinated
**Medication**	None	None	None	None
**Drug use**	Infrequent use of cigarettes and waterpipe; marijuana was tried 5 days prior to symptom onset; specific products consumed not determinable; other illicit drug use convincingly denied; negative urine toxicology screen	Frequent use of marijuana and waterpipe; specific products consumed not determinable; other illicit drug use convincingly denied	Smokes 10 cigarettes daily for 6 months, regular use of waterpipe and marijuana for several weeks; other illicit drug use denied; positive urine toxicology screen for cannabis	Regular use of tobacco and e-cigarettes, marijuana was smoked 2 and 3 days prior to admission
**Endoscopic findings**	Bronchial system shows no anatomic abnormalities. No inflammation of bronchial mucosa, little secretion. No granulomas, no morphologic correlate to a bronchiolitis, no signs of malignancy.	Bronchial system shows no anatomic abnormalities. Mildly inflamed bronchial system with foamy secretion. BAL: neutrophilic/eosinophilic inflammation in bronchi and alveoli.	/	BAL: nonspecific findings with 3% lymphocytes and no infectious pathogens
**Infection**	Moderately elevated *Mycoplasma pneumoniae* serology Negative respiratory PCR for *mycoplasma, legionella, mycobacteria, chlamydophila, CMV,* and *pneumocystis jirovecii*. Negative results for *HIV, Influenza viruses, Parainfluenza virus, RSV, EBV, CMV, HHV6, HSV, Adenovirus, Enterovirus*	Elevated *Chlamydophila pneumonia* serology Negative respiratory PCR for *chlamydophila, mycoplasma, legionella, mycobacteria, pneumocystis jirovecii, HSV, VZV, EBV, CMV, HHV6, Adenovirus, Influenza viruses, RSV, Parainfluenza virus,* and *Metapneumovirus*. Negative serologic tests for *aspergillus* or *candida*.	SARS-CoV-2: Negative pharyngeal swab SARS-CoV-2 PCR before admission and on days 0 and 4 after admission. Positive SARS-CoV-2 PCR on days 5 (C(t) > 35), 8 (C(t) = 22), 11 (C(t) = 18.3), 17, 21 (C(t) = 29.1) and 59. Negative SARS-CoV-2 serology on day 8. Further infectious work-up: Negative respiratory PCR for *chlamydophila, mycoplasma, legionella, bordetella, human rhinovirus/enterovirus, Adenovirus, Influenza viruses, RSV, Parainfluenza virus, coronavirus,* and *MERS-CoV*. Positive IgG serology for *Mycoplasma pneumoniae*, *Chlamydophila pneumoniae* Negative HIV-test. Negative blood cultures. Negative interferon-gamma release assay. Sputum negative for mycobacteria.	SARS-CoV-2: Negative pharyngeal swab SARS-CoV-2 PCR on admission Negative HIV-test. Negative blood cultures. Negative interferon-gamma release assay. Sputum negative for mycobacteria.
**Further results**	WES insignificant, no hypersensitivities in allergy panel; no cold-agglutinin antibodies, Coombs tests negative; slightly increased anti-CCP antibody (10.4 U/mL), otherwise insignificant rheumatologic results; insignificant immunological work-up.	Decreased IgG level (597 mg/dL), no IgG-subclass-irregularities. No WES performed. Inborn anisocoria (right > left), telangiectasia in pons and epidermoid cyst in cerebellopontine angle.	/	/
**Outpatient treatment**	Azithromycin, cefuroxime, 1× methylprednisolone burst (8 mg/kg for 3 days), inhaled formoterol/beclometason	Azithromycin, cefuroxime, 10× methylprednisolone burst (10 mg/kg for 3 days), inhaled tobramycin, salmeterol/fluticasone	Cefpodoxim, prednisolone (6 mg/kg for 3 days), inhaled salbutamol, ipratropiumbromid, budesonid	none

BAL = bronchoalveolar lavage, WES = whole-exome sequencing.

## Data Availability

The datasets used and/or analysed during the current study are available from the corresponding author on reasonable request. All data generated or analyzed during this study are included in this published article.

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
