# Peer review of "Do Not Miss Acute Diffuse Panbronchiolitis for Tree-in-Bud: Case Series of a Rare Lung Disease"

_diagnostics, 2022, doi:10.3390/diagnostics12071653_

Round 1

Reviewer 1 Report

The authors report the cases of 4 teenagers presenting with very similar cases of acute panbronchiolitis secondary to the voluntary inhalation of waterpipe and/or marijuana.

The authors highlight the lack of diagnosis on a subnormal chest radiography, the unique and typical CT-aspect with tree in bud pattern, and the favorable evolution in all 4 patients with corticosteroid treatment. This entity, rare and difficult to assess requires the patient’s confidence in admitting his non-legal consummations. It has to be distinguished from the Japanese panbronchiolitis.

The case series is very interesting, and well described.

I would suggest a few minor improvements:

-       Title: why introducing the ILD term whilst it is a bronchiolar disease? This point is not discussed anywhere in the text and I would suggest to skip this part of the title

-       Table 1: replace BAL by endoscopic findings and/or provide BAL results? Patient 2: how was the neutrophilic/eosinophilic inflammation assessed, bronchial biopsy? Too many details in this Table are repetitions from the text and could be deleted.

-       Case 2: Is it 10mg/kg/day or 10 bursts?

-       Figure 1: not sure if it provides any important information (CRP info could be added in the text as it is always<25; same for maximal pCO2 and time to normalization; same for FVC and FEV1). And it’s hard to read (too little).

-       Figure 2: major importance … increase the size of the CT images.

-       Discussion: the Japanese panbronchiolitis (or Asian) could be more detailed as this entity is also poorly known and deserves to be more descried in the text to better understand the differences. An “acute” may be missing Line 172 (just before the Asian sentence).

Author Response

Point 1:  Title: why introducing the ILD term whilst it is a bronchiolar disease? This point is not discussed anywhere in the text and I would suggest to skip this part of the title

Response to Point 1: Thank you very much for this important question. We added an explanation for our terminology and inserted a citation to a recent review article, which provides further information. Generally, we categorize the acute diffuse panbronchiolitis as an exposure related, non‐infectious diffuse parenchymal disorder (see: Griese, M. Etiologic Classification of Diffuse Parenchymal (Interstitial) Lung Diseases. Journal of Clinical Medicine 2022, 11, 1747, doi:10.3390/jcm11061747.). In addition to the respiratory bronchioles, the pulmonary interstitial space with the adjacent alveolar ducts and alveoli are affected. Differential diagnosis includes infectious bronchiolitis, aspiration bron-chiolitis, exogenous allergic alveolitis, granulomatous polyangiits, bronchocentric granulomatosis and malignant lymphoma.

Point 2:    Table 1: replace BAL by endoscopic findings and/or provide BAL results? Patient 2: how was the neutrophilic/eosinophilic inflammation assessed, bronchial biopsy? Too many details in this Table are repetitions from the text and could be deleted.

Response to Point 2: Thanks for these suggestions, which we implemented: in Table 1 we have changed the line heading to „Endoscopic findings / Bronchoalveolar lavage (BAL)“. Repetitions from the text and table have been deleted. In patient 2 the neutrophilic/eosinophilic inflammation was assessed by BAL, therefore we specified this in the table.

Point 3: Case 2: Is it 10mg/kg/day or 10 bursts?

Response to Point 3: Thanks for the question: in total the patient received 10 rounds of methylprednisolone bursts with 10mg/kg for 3 days. We have specified this information in Figure 1.

Point 4: Figure 1: not sure if it provides any important information (CRP info could be added in the text as it is always<25; same for maximal pCO2 and time to normalization; same for FVC and FEV1). And it’s hard to read (too little).

Response to Point 4: Figure 1: we think the progression of CRP and maximal pCO2 as well as pulmonary function test results are of interest and can be most easily displayed visually. We added total lung capacity values as % of predicted values and increased the text size.

Point 5: Figure 2: major importance … increase the size of the CT images.

Response to Point 5: Figure 2: to increase the size of the CT images we had to split the figure in two separate figures (Figure 2: Patient 1+2; Figure 3: Patient 3+4)

Point 6: Discussion: the Japanese panbronchiolitis (or Asian) could be more detailed as this entity is also poorly known and deserves to be more descried in the text to better understand the differences. An “acute” may be missing Line 172 (just before the Asian sentence).

Response to Point 6: Thank you for this input. We added more details regarding the Japanese panbronchiolitis to better differentiate the disease from our cases (Line 205-210). An “acute” was added in Line 172.

Reviewer 2 Report

The article by Raedler and colleagues: “ Don’t miss acute diffuse panbronchiolitis for the tree in bud – case series of a rare interstitial lung disease” is a very interesting report on bronchiolitis in patients who smoked waterpipe, tobacco or cannabinoids. The disease pattern is different from the one observed in patients with EVALI.  

There are some issues that authors need to address.

1.     My greatest concern would be how the authors can be sure that the changes are caused solely by the inhalation of the noxious agents and not the infectious agents alone or acting together with inhaled fumes. CRP was slightly increased in all the patient initially and in patient 2 it was fluctuating throughout the follow-up periodTwo patients had serological evidence of past infection with atypical bacteria, one had active SARS-Cov-2 infection and patient 4 had not been tested for respiratory pathogens (apart from SARS Cov-2 and Mycobacteria). There were also decreased IgG levels detected in Patient 2 without further investigation towards immune deficiency syndromes (eg. Commencing CVID) that could be partly responsible for changes in the lungs.

2.     It would be interesting to see pulmonary function tests results of these patients esp. TLC and RV/TLC from Bodyplethysmography and TLCO

Author Response

Point 1: My greatest concern would be how the authors can be sure that the changes are caused solely by the inhalation of the noxious agents and not the infectious agents alone or acting together with inhaled fumes. CRP was slightly increased in all the patient initially and in patient 2 it was fluctuating throughout the follow-up periodTwo patients had serological evidence of past infection with atypical bacteria, one had active SARS-Cov-2 infection and patient 4 had not been tested for respiratory pathogens (apart from SARS Cov-2 and Mycobacteria). There were also decreased IgG levels detected in Patient 2 without further investigation towards immune deficiency syndromes (eg. Commencing CVID) that could be partly responsible for changes in the lungs.

Response to Point 1: It is absolutey right, that we can not be sure, that changes are caused solely by the inhalation of the noxious agents. We acknowledge that fact in Line 225-226 with the following sentence: „Possibly a combination of as yet unknown precipitating factors, such as an undetected infection or toxins, may be causative in our patients.”  

Point 2: It would be interesting to see pulmonary function tests results of these patients esp. TLC and RV/TLC from Bodyplethysmography and TLCO

Response to Point 2: We totally agree, it would be desirable to have some more pulmonary function tests from our patients, especially from patient 3 + 4. For patients 1 and 2 we inserted TLC in Figure 1. Bodyplethysmographiy was only performed once during follow-up of Patient 2 after 18 months. We added the information in the manuscript (Line 145-147). Due to the patient´s age, Patient 1 and 2 were transitioned to adult care and we have lost follow-up. Unfortunately, patient 3 lacks compliance for further diagnostic tests including pulmonary function tests, because he subjectively „feels good by now“. Patient 4 was only recently diagnosed, therefore there is no longer follow-up available, by now.